# Microwave Assisted Reactions of Fluorescent Pyrrolodiazine Building Blocks

**DOI:** 10.3390/molecules24203760

**Published:** 2019-10-18

**Authors:** Costel Moldoveanu, Dorina Amariucai-Mantu, Violeta Mangalagiu, Vasilichia Antoci, Dan Maftei, Ionel I. Mangalagiu, Gheorghita Zbancioc

**Affiliations:** 1Chemistry Department, Alexandru Ioan Cuza University of Iasi, 11 Carol 1st Bvd, 700506 Iasi, Romania; dorina.mantu@uaic.ro (D.A.-M.); vasilichia.antoci@uaic.ro (V.A.); dan.maftei@chem.uaic.ro (D.M.); ionelm@uaic.ro (I.I.M.); 2Institute of Interdisciplinary Research CERNESIM Centre, Alexandru Ioan Cuza University of Iasi, 11 Carol I, 700506 Iasi, Romania; violeta.mangalagiu@uaic.ro

**Keywords:** pyrrolopyridazine, pyrrolophthalazine, microwave irradiation, fluorescent

## Abstract

We report here the synthesis and optical spectral properties of several new pyrrolodiazine derivatives. The luminescent heterocycles were synthesized by 1,3-dipolar cycloaddition reactions between N-alkylated pyridazine and methylpropiolate or dimethyl acetylenedicarboxylate (DMAD). The pyrrolopyridazine derivatives are blue emitters with moderate quantum yields (around 25%) in the case of pyrrolopyridazines and negligible yet measurable emission for pyrrolophthalazines. In a subsequent step towards including the pyrrolodiazine moiety, given its spectral properties in various macromolecular frameworks such as biological molecules, a subset of the synthetized compounds has been subjected to α-bromination. A selective and efficient way for α-bromination in heterogeneous catalysis of pyrrolodiazine derivatives under microwave (MW) irradiation is presented. We report substantially higher yields under MW irradiation, whereas the solvent amounts required are at least five-fold less compared to classical heating.

## 1. Introduction

Pyrrolodiazines are an important class of N-bridgehead heterocycles which have received increasing interest during the last few years driven by a wide range of potential applications, from electroluminescent materials [1] to interesting biological uses [2]. In recent years, azaheterocycle derivatives have been reported to display a large variety of applications in the fields of medicinal chemistry (these include antimicrobials, anti-cancer, antioxidant, anti-human immunodeficiency virus (HIV), etc.) [2,3,4,5,6,7,8]. The interest in the field of pyrrolodiazine derivatives arises also from their highly efficient blue fluorescence emission [9,10,11]. The latter property makes pyrrolodiazine derivatives very attractive materials in optoelectronics for blue organic light-emitting diodes [9,10,11], whereas a combined use of the two distinct properties has suggested interesting applications as fluorescent biomarkers [12,13,14].

Pyrrolodiazines offer very interesting optical properties. Azaindolizines derivatives represent such a class (with 10 π-electron N-fused heterocycles, containing a bridgehead nitrogen atom shared by an electron-excessive pyrrole and a diazine electron deficient six-membered ring) being a ‘pure’ blue-emitting moiety. [15,16]. This uneven π-electron distribution between the two fused rings is an important feature that leads to electron delocalization. The electron delocalization within the entire heterocycle skeleton can be possible without a planar geometry of the indolizine. The planarity of azaindolizine is provided by the sp^2^ hybridization of all the atoms in the fused ring and is preserved upon substitution with different groups.

As a relatively new trend in synthetic organic chemistry, microwave (MW)-assisted reactions offer a versatile and facile pathway for an increasing number of organic syntheses [17,18]. In this respect, our group has presented (in previous work) several studies on microwave-assisted dipolar cycloaddition reactions of diazinium ylides with various dipolarophiles. Herein, we add further contributions to the field of using MW irradiation in organic chemistry [19,20,21,22,23,24,25], and propose a facile, efficient and environmentally friendly method for the synthesis of pyrrolodiazine derivatives using MW technologies.

## 2. Results and Discussion

The general approach adopted for the synthesis of fluorescent pyrrolodiazine derivatives is depicted in Scheme 1. As shown below, the preparation of all pyrrolodiazine derivatives, **7–11**, involves two steps: initially N-alkylation of the diazine (pyridazine—PY or phthalazine—PH) with bromoacetone **3** followed by a 3 + 2 dipolar cycloaddition of diazinium ylides **6** (generated in situ in the presence of triehtylamine—TEA) from the corresponding salts to the corresponding dipolarophiles (dimethyl acetylenedicarboxylate—DMAD or methyl propiolate). Bromoacetone was synthesized, in a preliminary step, by the reaction of acetone with bromine in acetic acid as a catalyst.

An alternative of this synthetic approach was used by Masaki and coworkers when investigating the cycloaddition reaction of pyridazinium ylides [26]. They performed the reaction in dimethylformamide (DMF) or benzene in the presence of TEA (in order to generate the ylide), the intermediate products being then oxidized with chloranil to obtain the fully aromatized cycloadducts **7** and **8** from Scheme 1. The structure of the resulting pyrrolo[1,2-b]pyridazines has been determined by elemental and spectral analysis. Since the NMR data is presented in the work referenced above in a rather outdated manner (i.e., chemical shift was presented in τ and not in δ, which is more actual), we chose to present a thorough spectral characterization of the compounds **7** and **8** in the experimental part of this work.

Another alternative to the synthetic approach was used by Dumitrascu and coworkers [27] in order to obtain pyrrolo[2,1-a]phthalazines. They performed the reaction at low temperature (ice-cooled) in the presence of TEA but obtained only the hydrogenated cycloadduct **11** from Scheme 1. In subsequent studies, Dumitrascu and coworkers [28] also obtained fully aromatized cycloadducts of type **9** and **10**, but they use 1,2-epoxybutane (instead of TEA) which has the role of a solvent, to generate the corresponding ylide from the phthalazinium chlorides. However, in the two subsequent works, the authors were interested in the synthesis and elucidation of the molecular geometry and crystalline structure and did not approach the photophysical properties of these compounds.

In the current work, we performed the reactions in refluxing chloroform, and a chloroform solution of TEA was added dropwise at the beginning of the reaction. After 6 h of reflux, the reaction was stopped and the products were isolated from the reaction mixture. In the case of pyridazine, we only obtained the fully aromatized compounds **7** and **8**. For phthalazine, the reaction with methyl propiolate yields the fully aromatized compound **9**, while the reaction with DMAD gives a mixture of partial and fully aromatized compounds (**11** and **10**, respectively). All cycloadducts were obtained in moderate to good yield (74% to 81%, see Table 1). The results obtained by us in this study are in quantitative agreement with those published in the literature, including on the geometry of the hydrogenated cycloadduct **11**. We note, however, besides the high energy consumption, the long reaction time (360 min) is major disadvantage of the synthesis carried out under conventional conditions.

As an alternative route, we have performed the synthesis of the azaindolizine derivatives under MW irradiation, using a monomode reactor (Monowave 300; Anton Paar, Graz, Austria). This reactor is equipped with a stirring system (0 to 1200 rpm) and can reach up to 300 °C with temperature control. The reactions take place in a closed vessel at 30 bars of maximum pressure. The optimal reaction conditions were found to be at 155 °C and 16–17 bars, and are summarized in Table 1 compared to the corresponding parameters used in thermal heating (TH) conditions.

As indicated in Table 1, under MW irradiation, the reaction times decrease substantially from 6 h to 10 min, whereas the solvent amounts used in the former were at least five times lower than the corresponding quantities used under conventional conditions (see Experimental section). This qualifies the former reactions as environmentally friendly. We also noticed that under MW irradiation, the yields were higher (by 10% to 15%). In the case of the reaction of phthalazinium bromide with DMAD, under microwave irradiation, we isolated only fully aromatized cycloadduct **10** with a >91% yield.

Photophysical properties of the synthetized pyrrolodiazines were investigated using diluted solutions (less than 1 × 10^−5^ mol/L) prepared in cyclohexane and dichloromethane, respectively. The dilution of each solution was adjusted and thus the absorbances, measured at 360 nm, 320 nm and 370 nm and reported on a 10 mm cuvette, to fit in the range of 0.02–0.20 units.

The optical absorption and emission maxima with corresponding quantum yield of the pyrrolodiazine derivatives in cyclohexane and dichloromethane are summarized in Table 2.

Table 2 shows that compounds **7** and **8** are moderate blue emitters (λ_max_ of fluorescence around 420–430 nm, with quantum yields around 25%), compounds **9** and **10** are still blue emitters (λ_max_ of fluorescence around 425–440 nm, with lower quantum yields—around 5%), while for compound **11**, the fluorescence intensity is negligible and the emission is substantially red shifted, with a maximum of the emission band (measurable only in cyclohexane) around 490 nm. Such a deviation from the trends shown in Table 2 should relate to the difference in the electronic structures of **7**–**10** (aromatic and fully conjugated), and **11** that lacks a full electron conjugation.

It is the fluorescence emission of the fully aromatized compounds such as **7**–**10** that justifies the interest in synthesizing pyrrolodiazines. However, for most of the anticipated applications, including their use as fluorescent markers, choosing a feasible derivatization reaction is important in order to incorporate them into biological macromolecules (peptides, proteins and DNA). Hence, taking into account that the carbonyl group can be easily halogenated in the α-position, and that the resulting derivatives should present an enhanced reactivity, we decided in the next step to investigate the bromination reaction of pyrrolodiazines **7**–**10** (shown in Scheme 2). The procedure chosen has been reported previously by us in [29], and consists of bromination in heterogeneous catalysis using copper (II) bromide in chloroform/ethyl acetate. This reaction system is highly regioselective, leading to α-bromo derivatives in good yields, but less selective regarding to the resulting mono- or di-brominated product. Using this procedure, we performed the bromination of compounds **7**–**10** with copper (II) bromide, both under conventional TH and MW irradiation, in order to study the reaction selectivity. In Table 3, we list the optimized conditions that were employed.

As shown in Table 3, the microwave-assisted bromination of pyrrolodiazine derivatives **7**–**10** occurred with the expected regioselectivity (only in the α-position) and with increased selectivity regarding to the monobrominated compound. The ratio between mono- and di-brominated compounds was around 3:1 under thermal heating, and 6:1 under unconventional heating. The yields were moderate under conventional TH and good under MW irradiation, which showed a significant increase of yields (an average of 15–20%).

## 3. Experimental Section

### 3.1. General Procedure

All the reagents and solvents were purchased from commercial sources and used without further purification (except bromoacetone, which was synthesized by the reaction of acetone with bromine in acetic acid as a catalyst). Melting points were recorded on an Electrothermal MEL-TEMP (Barnstead International, Dubuque, IA, USA) apparatus in open capillary tubes and are uncorrected. Analytical thin-layer chromatography was performed with commercial silica gel plates 60 F254 (Merck Darmstadt, Germany) and visualized with UV light. The NMR spectra were recorded on an Avance III 500 MHz spectrometer (Bruker, Vienna, Austria) operating at 500 MHz for ^1^H and 125 MHz for ^13^C. The following abbreviations were used to designate chemical shift multiplicities: s = singlet, d = doublet, t = triplet, m = multiplet. Chemical shifts were reported in delta (δ) units, part per million (ppm) and coupling constants (*J*) in Hz. Infrared (IR) data were recorded as films on potassium bromide (KBr) pellets on a FT-IR Prestige 8400s spectrophotometer (Shimadzu, Kyoto, Japan). For the microwave irradiation, we used a monomode reactor (Monowave 300; Anton Paar, Graz, Austria). This reactor can reach up to 300 °C and controls the temperature via a built-in IR sensor. Some additional specifications of the Monowave 300 include: microwave power = 850 W, operation limits at 300 °C and 30 bars, reaction vial = borosilicate, operation volume = between 0.5–20 mL, pressure control by hydraulic system, agitation with an integrated magnetic stirrer (0 to 1200 rpm) and cooling with compressed air. UV-Vis spectra were recorded on a Shimadzu 1800 PC spectrophotometer in cyclohexane and dichloromethane (spectroscopic grade) solution. The fluorescence measurements were made using a F900 photoluminescence spectrometer (Edinburgh Instruments, Livingstone, UK) in the same solvents as for the UV-Vis spectra, with the excitation wavelength set to the absorption band maximum. For all spectral determinations, the solutions were kept in 10 mm path length quartz cells. The fluorescence quantum yield was determined at room temperature with a FLS 980 fluorimeter (Edinburgh Instruments) with an integrating sphere and the excitation wavelength corresponds to the maximum of the absorption band.

#### 3.1.1. General Procedure for Synthesis of Pyrrolodiazine Derivatives **7**–**11** under Conventional TH Conditions and MW Irradiation

A mixture of diazinium salt **4** or **5** (0.543 g, 2.5 mmol for pyridazinium bromide **4** or 0.668 g, 2.5 mmol for phthalazinium bromide **5**) and dimethyl acetylenedicarboxylate (0.43 mL, 3.5 mmol) or methyl propiolate (0.31 mL, 3.5 mmol) was suspended in 35 mL chloroform. Then, triethylamine (0.48 mL, 3.5 mmol) dissolved in 15 mL chloroform was added dropwise under stirring and refluxing in one hour. The stirring and refluxing were continued for 360 min. After the reaction was finished (TLC), the obtained solution was cooled down to room temperature and then the reaction mixtures were washed with water (3 × 25 mL), dried over magnesium sulfate and evaporated under reduced pressure to give the crude product. The purification of the crude product was done by column chromatography on silica gel (eluted with 99.5/0.5 CH_2_Cl_2_/CH_3_OH).

Under MW irradiation, the mixture of reagents (in 10 mL chloroform) was placed into the reaction vessel and exposed to irradiation for 10 min. Using MW irradiation, the best results were obtained using a “temperature control” method. The “temperature control” method ensures a constant temperature (in this case 100 °C) to vary the magnetron power. This method takes place in three stages. In the first step, the temperature is raised as quickly as possible (within less than 1 min) by applying maximum power. In the second step, the reaction mixture is kept at a constant temperature with the control of the magnetron power. In the last step, the reaction tube is cooled to 55 °C by stopping the irradiation and blowing the reaction vial with compressed air. Once the heating cycle is completed, the reaction vial is removed from the reactor and processed as indicated for TH. The NMR spectra and IR Spectra of the obtained compounds are in the Appendix A.

#### 3.1.2. General Procedure for Synthesis of α-brominated Pyrrolodiazines **12**–**15** under Conventional TH Conditions and MW Irradiation

The pyrrolodiazine derivatives **7**–**10** (1 mmol, 0.218 g for cycloadduct **7**; 0.276 g for cycloadduct **8**; 0.268 g for cycloadduct **9**; 0.326 g for cycloadduct **10**) dissolved in 10 mL of a mixture of chloroform/ethyl acetate, were added dropwise under stirring and refluxing in one hour to a suspension of copper (II) bromide (2 mmol, 0.448 g) in a 20 mL mixture of chloroform/ethyl acetate at a 1:2 ratio (pyrrolodiazine derivatives **7**–**10** / copper (II) bromide). The stirring and refluxing were continued for 480 min. The hot solution was filtered in order to remove the copper (I) bromide that formed. The solvent was evaporated by vacuum distillation. The crude product was purified by column chromatography on a silica gel (eluted with dichloromethane).

Under MW irradiation, the pyrrolodiazine derivatives **7**–**10** (1 mmol, 0.218 g for cycloadduct **7**; 0.276 g for cycloadduct **8**; 0.268 g for cycloadduct **9**; 0.326 g for cycloadduct **10**) and copper (II) bromide (2 mmol, 0.448 g) were mixed at a 1:2 ratio (pyrrolodiazine derivatives **7–10**/copper (II) bromide) in 10 mL solvent (chloroform/ethylacetate). This mixture was then placed in the reaction vessel and exposed to irradiation for 20 min. Using MW irradiation, the best results were obtained using a “temperature control” method. The “temperature control” method ensures a constant temperature (in this case 120 °C) while varying the magnetron power. Once the heating cycle was completed, the reaction tube was cooled to ambient temperature, removed from the reactor, and processed as indicated for TH.

*Methyl 7-acetylpyrrolo[1,2-b]pyridazine-5-carboxylate* (**7**). 0.42 g, 77% (under classical heating) and 0.49 g, 90% (under microwave) as white crystals, m.p. 138–140 °C; *R*_f_ (98/2 CH_2_Cl_2_/CH_3_OH) 0.37; IR (cm^−1^): 3111, 3103, 3066 (C-H arom.), 2953 (C-H aliph.), 1701 (C=O, ester), 1659 (C=O, keto), 1616, 1529, 1519, 1448, 1379 (aromatic and heteroaromatic ring), 1282, 1249, 1211, 1176, 1122 (C–O–C, ester); ^1^H NMR (500 MHz, CDCl_3_): δ 8.65 (1H, dd, *J* = 2.0, 9.0 Hz, H-2), 8.51 (1H, dd, *J* = 2.0, 4.0 Hz, H-4), 8.02 (1H, s, H-6), 7.12 (1H, dd, *J* = 4.0, 9.0 Hz, H-3), 3.93 (3H, s, CH_3_ of methoxycarbonyl group from 5 position), 2.71 (3H, s, CH_3_ of acetyl group from 7 position); ^13^C NMR (125 MHz, CDCl_3_): δ 187.4 (CO keto group from 7 position), 164.0 (CO keto ester from 5 position), 144.2 (C-4), 133.2 (C-4a), 128.2 (C-2), 128.1 (C-7), 123.0 (C-6), 117.3 (C-3), 105.4 (C-5), 51.7 (CH_3_ of methoxycarbonyl group from 5 position), 29.5 (CH_3_ of acetyl group from 7 position). All spectral data are in agreement with the previously reported data [26].

*Dimethyl 7-acetylpyrrolo[1,2-b]pyridazine-5,6-dicarboxylate* (**8**). 0.51 g, 74% (under classical heating) and 0.63 g, 91% (under microwaves) as white crystals, m.p. 197–198 °C; *R*_f_ (98/2 CH_2_Cl_2_/CH_3_OH) 0.38; IR (cm^−1^): 3082, 3064, 3010 (C-H arom.), 2957 (C-H aliph.), 1743, 1705 (C=O, ester), 1660 (C=O, keto), 1614, 1535, 1500, 1452, 1381 (aromatic and heteroaromatic ring), 1313, 1269, 1244, 1211, 1168, 1130, 1072 (C–O–C, ester); ^1^H NMR (500 MHz, CDCl_3_): δ 8.65 (1H, dd, *J* = 2.0, 9.0 Hz, H-2), 8.46 (1H, dd, *J* = 2.0, 4.5 Hz, H-4), 7.14 (1H, dd, *J* = 4.5, 9.0 Hz, H-3), 4.02 (3H, s, CH_3_ of methoxycarbonyl group from 5 position), 3.92 (3H, s, CH_3_ of methoxycarbonyl group from 6 position), 2.81 (3H, s, CH_3_ of acetyl group from 7 position); ^13^C NMR (125 MHz, CDCl_3_): δ 187.7 (CO keto group from 7 position), 166.0 (CO keto ester from 6 position), 162.8 (CO keto ester from 5 position), 144.3 (C-4), 131.3 (C-4a), 128.8 (C-2), 128.1 (C-7), 126.8 (C-6), 117.3 (C-3), 103.4 (C-5), 53.1 (CH_3_ of methoxycarbonyl group from 6 position), 51.9 (CH_3_ of methoxycarbonyl group from 5 position), 31.0 (CH_3_ of acetyl group from 7 position). All spectral data are in agreement with the previously reported data [26].

*Methyl 3-Aetylpyrrolo[2,1-a]phthalazine-1-carboxylate* (**9**). 0.54 g, 81% (under classical heating) and 0.63 g, 94% (under microwaves) as white crystals, m.p. 194–195 °C; *R*_f_ (98/2 CH_2_Cl_2_/CH_3_OH) 0.38. All spectral data are in agreement with the previously reported data [28].

*Dimethyl 3-acetylpyrrolo[2,1-a]phthalazine-1,2-dicarboxylate* (**10**). 0.37 g, 45% (under classical heating) and 0.74 g, 91% (under microwaves) as white crystals, m.p. 148–150 °C; *R*_f_ (98/2 CH_2_Cl_2_/CH_3_OH) 0.40. All spectral data are in agreement with the previously reported one [28].

*Dimethyl 3-acetyl-1,10b-dihydropyrrolo[2,1-a]phthalazine-1,2-dicarboxylate* (**11**). 0.30 g, 37% (under classical heating) and 0.0 g, 0% (under microwaves) as yellow crystals, m.p. 138–139 °C; *R*_f_ (98/2 CH_2_Cl_2_/CH_3_OH) 0.50. All spectral data are in agreement with the previously reported data [27].

*Methyl 7-(2-bromoacetyl)pyrrolo[1,2-b]pyridazine-5-carboxylate* (**12a**). 0.16 g, 53% (under classical heating) and 0.23 g, 77% (under microwaves) as yellowish crystals, m.p. 140–141 °C; *R*_f_ (99/1 CH_2_Cl_2_/CH_3_OH) 0.33; IR (cm^−1^): 3090, 3074, 3010 (C-H arom.), 2955, 2924 (C-H aliph.), 1707 (C=O, ester), 1654 (C=O, keto), 1529, 1471, 1450, 1379 (aromatic and heteroaromatic ring), 1286, 1263, 1226, 1197, 1136 (C–O–C, ester), 582 (C–Br); ^1^H NMR (500 MHz, CDCl_3_): δ 8.66 (1H, dd, *J* = 1.5, 9.0 Hz, H-2), 8.51 (1H, dd, *J* = 1.5, 4.5 Hz, H-4), 8.11 (1H, s, H-6), 7.16 (1H, dd, *J* = 4.5, 9.0 Hz, H-3), 4.64 (2H, s, CH_2_ of bromoacetyl group from 7 position), 3.92 (3H, s, CH_3_ of methoxycarbonyl group from 5 position); ^13^C NMR (125 MHz, CDCl_3_): δ 181.0 (CO keto group from 7 position), 163.7 (CO keto ester from 5 position), 144.3 (C-4), 133.5 (C-4a), 128.4 (C-2), 126.0 (C-7), 123.6 (C-6), 117.6 (C-3), 106.4 (C-5), 51.8 (CH_3_ of methoxycarbonyl group from 5 position), 34.2 (CH_2_ of bromoacetyl group from 7 position).

*Methyl 7-(2,2-dibromoacetyl)pyrrolo[1,2-b]pyridazine-5-carboxylate* (**12b**). 0.064 g, 17% (under classical heating) and 0.045 g, 12% (under microwaves) as yellowish crystals, m.p. 150–151 °C; *R*_f_ (99/1 CH_2_Cl_2_/CH_3_OH) 0.58; IR (cm^−1^): 3095, 3061, 3049, 3034 (C-H arom.), 2945 (C-H aliph.), 1710 (C=O, ester), 1666 (C=O, keto), 1535, 1518, 1467, 1442, 1415 (aromatic and heteroaromatic ring), 1253, 1234, 1203, 1165, 1138 (C–O–C, ester), 663, 621 (C–Br); ^1^H NMR (500 MHz, CDCl_3_): δ 8.70 (1H, dd, *J* = 2.0, 9.0 Hz, H-2), 8.55 (1H, dd, *J* = 2.0, 4.5 Hz, H-4), 8.27 (1H, s, H-6), 7.35 (1H, s, CH of dibromoacetyl group from 7 position), 7.20 (1H, dd, *J* = 4.5, 9.0 Hz, H-3), 3.94 (3H, s, CH_3_ of methoxycarbonyl group from 5 position); ^13^C NMR (125 MHz, CDCl_3_): δ 176.2 (CO keto group from 7 position), 163.6 (CO keto ester from 5 position), 144.4 (C-4), 134.7 (C-4a), 134.1 (C-7), 128.7 (C-2), 124.9 (C-6), 117.8 (C-3), 107.2 (C-5), 51.9 (CH_3_ of methoxycarbonyl group from 5 position), 42.5 (CH of dibromoacetyl group from 7 position).

*Methyl 3-(2-bromoacetyl)pyrrolo[2,1-a]phthalazine-1-carboxylate* (**13a**). 0.17 g, 50% (under classical heating) and 0.26 g, 76% (under microwaves) as yellowish crystals, m.p. 155–156 °C; *R*_f_ (99/1 CH_2_Cl_2_/CH_3_OH) 0.51; IR (cm^−1^): 3115, 3039, 3020 (C-H arom.), 2949, 2904 (C-H aliph.), 1703 (C=O, ester), 1662 (C=O, keto), 1554, 1529, 1487, 1450, 1375 (aromatic and heteroaromatic ring), 1255, 1188, 1172, 1143, 1089 (C–O–C, ester), 609 (C–Br); ^1^H NMR (500 MHz, CDCl_3_): δ 9.82 (1H, d, *J* = 8.5 Hz, H-10), 8.75 (1H, s, H-6), 8.14 (1H, s, H-2), 7.90-7.94 (2H, m, H-7, H-9), 7.77 (1H, dd, *J* = 7.0, 8.0 Hz, H-8), 4.72 (2H, s, CH_2_ of bromoacetyl group from 3 position), 3.97 (3H, s, CH_3_ of methoxycarbonyl group from 1 position); ^13^C NMR (125 MHz, CDCl_3_): δ 181.6 (CO keto group from 3 position), 164.6 (CO keto ester from 1 position), 146.5 (C-6), 133.5 (C-9), 130.3 (C-10b), 130.2 (C-8), 127.8 (C-7),127.7 (C-10), 126.9 (C-10a), 126.3 (C-3), 124.0 (C-2), 121.9 (C-6a), 109.3 (C-1), 52.2 (CH_3_ of methoxycarbonyl group from 1 position), 34.8 (CH_2_ of bromoacetyl group from 3 position).

*Methyl 3-(2,2-dibromoacetyl)pyrrolo[2,1-a]phthalazine-1-carboxylate* (**13b**). 0.068 g, 16% (under classical heating) and 0.047 g, 11% (under microwaves) as yellowish crystals, m.p. 192–193 °C; *R*_f_ (99/1 CH_2_Cl_2_/CH_3_OH) 0.72; IR (cm^−1^): 3113, 3043, 3018, 2997 (C-H arom.), 2947(C-H aliph.), 1720 (C=O, ester), 1678 (C=O, keto), 1531, 1462, 1448, 1404, 1379 (aromatic and heteroaromatic ring), 1265, 1240, 1178, 1143 (C–O–C, ester), 663, 624 (C–Br); ^1^H NMR (500 MHz, CDCl_3_): δ 9.81 (1H, d, *J* = 8.0 Hz, H-10), 8.78 (1H, s, H-6), 8.26 (1H, s, H-2), 7.92-7.96 (2H, m, H-7, H-9), 7.80 (1H, dd, *J* = 7.0, 7.5 Hz, H-8), 7.49 (1H, s, CH of dibromoacetyl group from 3 position), 3.98 (3H, s, CH_3_ of methoxycarbonyl group from 1 position); ^13^C NMR (125 MHz, CDCl_3_): δ 176.7 (CO keto group from 3 position), 164.4 (CO keto ester from 1 position), 146.6 (C-6), 133.7 (C-9), 130.9 (C-10b), 130.4 (C-8), 127.9 (C-7),127.9 (C-10), 126.9 (C-10a), 125.5 (C-2), 122.8 (C-3), 121.9 (C-6a), 110.0 (C-1), 52.2 (CH_3_ of methoxycarbonyl group from 1 position), 43.1 (CH of dibromoacetyl group from 3 position).

*Dimethyl 7-(2-bromoacetyl)pyrrolo[1,2-b]pyridazine-5,6-dicarboxylate* (**14a**). 0.20 g, 55% (under classical heating) and 0.27 g, 77% (under microwaves) as yellowish crystals, m.p. 177–178 °C; *R*_f_ (99/1 CH_2_Cl_2_/CH_3_OH) 0.28; IR (cm^−1^): 3132, 3099, 3001 (C-H arom.), 2951, 2924 (C-H aliph.), 1749, 1714 (C=O, ester), 1660 (C=O, keto), 1502, 1450, 1427, 1381 (aromatic and heteroaromatic ring), 1249, 1228, 1190, 1170, 1116 (C–O–C, ester), 580 (C–Br); ^1^H NMR (500 MHz, DMSO): δ 8.83 (1H, dd, *J* = 1.5, 4.5 Hz, H-2), 8.65 (1H, dd, *J* = 1.5, 9.0 Hz, H-4), 7.50 (1H, dd, *J* = 4.5, 9.0 Hz, H-3), 5.02 (2H, s, CH_2_ of bromoacetyl group from 7 position), 3.88 (3H, s, CH_3_ of methoxycarbonyl group from 5 position), 3.85 (3H, s, CH_3_ of methoxycarbonyl group from 6 position); ^13^C NMR (125 MHz, DMSO): δ 180.6 (CO keto group from 7 position), 164.8 (CO keto ester from 6 position), 161.9 (CO keto ester from 5 position), 145.8 (C-4), 131.1 (C-4a), 128.5 (C-2), 128.4 (C-7), 123.6 (C-6), 119.4 (C-3), 102.9 (C-5), 52.6 (CH_3_ of methoxycarbonyl group from 6 position), 51.9 (CH_3_ of methoxycarbonyl group from 5 position), 37.5 (CH_2_ of bromoacetyl group from 7 position).

*Dimethyl 7-(2,2-dibromoacetyl)pyrrolo[1,2-b]pyridazine-5,6-dicarboxylate* (**14b**). 0.074 g, 17% (under classical heating) and 0.056 g, 13% (under microwaves) as yellowish crystals, m.p. 181–182 °C; *R*_f_ (99/1 CH_2_Cl_2_/CH_3_OH) 0.33; IR (cm^−1^): 3099, 3051, 3024 (C-H arom.), 2953 (C-H aliph.), 1735, 1712 (C=O, ester), 1656 (C=O, keto), 1535, 1502, 1440, 1423, 1390 (aromatic and heteroaromatic ring), 1271, 1251, 1224, 1203, 1176, 1114 (C–O–C, ester), 590, 547 (C–Br); ^1^H NMR (500 MHz, CDCl_3_): δ 8.72 (1H, dd, *J* = 1.5, 9.5 Hz, H-2), 8.57 (1H, dd, *J* = 1.5, 4.5 Hz, H-4), 7.62 (1H, s, CH of dibromoacetyl group from 7 position), 7.25 (1H, dd, *J* = 4.5, 9.5 Hz, H-3), 4.05 (3H, s, CH_3_ of methoxycarbonyl group from 5 position), 3.92 (3H, s, CH_3_ of methoxycarbonyl group from 6 position); ^13^C NMR (125 MHz, CDCl_3_): δ 176.0 (CO keto group from 7 position), 165.1 (CO keto ester from 6 position), 162.4 (CO keto ester from 5 position), 145.0 (C-4), 132.5 (C-4a), 131.4 (C-7), 129.4 (C-2), 120.5 (C-6), 118.3 (C-3), 105.2 (C-5), 53.5 (CH_3_ of methoxycarbonyl group from 6 position), 52.3 (CH_3_ of methoxycarbonyl group from 5 position), 43.0 (CH of dibromoacetyl group from 7 position).

*Dimethyl 3-(2-bromoacetyl)pyrrolo[2,1-a]phthalazine-1,2-dicarboxylate* (**15a**). 0.20 g, 49% (under classical heating) and 0.30 g, 75% (under microwaves) as yellowish crystals, m.p. 159–160 °C; *R*_f_ (99/1 CH_2_Cl_2_/CH_3_OH) 0.42; IR (cm^−1^): 3119, 3045, 3026, 2995 (C-H arom.), 2949 (C-H aliph.), 1722 (C=O, ester), 1664 (C=O, keto), 1552, 1527, 1500, 1467, 1388 (aromatic and heteroaromatic ring), 1259, 1234, 1195, 1168, 1114 (C–O–C, ester), 611 (C–Br); ^1^H NMR (500 MHz, CDCl_3_): δ 9.53 (1H, d, *J* = 8.5 Hz, H-10), 8.73 (1H, s, H-6), 7.90-7.93 (2H, m, H-7, H-9), 7.78 (1H, dd, *J* = 7.0, 7.5 Hz, H-8), 4.85 (2H, s, CH_2_ of bromoacetyl group from 3 position), 4.02 (3H, s, CH_3_ of methoxycarbonyl group from 1 position), 3.94 (3H, s, CH_3_ of methoxycarbonyl group from 2 position); ^13^C NMR (125 MHz, CDCl_3_): δ 181.7 (CO keto group from 3 position), 166.0 (CO keto ester from 2 position), 163.7 (CO keto ester from 1 position), 147.0 (C-6), 133.9 (C-9), 130.4 (C-8), 128.7 (C-3), 128.3 (C-10b), 128.2 (C-7), 127.3 (C-10), 126.7 (C-10a), 124.7 (C-2), 121.8 (C-6a), 107.6 (C-1), 53.2 (CH_3_ of methoxycarbonyl group from 2 position), 52.5 (CH_3_ of methoxycarbonyl group from 1 position), 36.6 (CH_2_ of bromoacetyl group from 3 position).

*Dimethyl 3-(2,2-dibromoacetyl)pyrrolo[2,1-a]phthalazine-1,2-dicarboxylate* (**15b**). 0.077 g, 16% (under classical heating) and 0.058 g, 12% (under microwaves) as yellowish crystals, m.p. 181–182 °C; *R*_f_ (99.9/0.1 CH_2_Cl_2_/CH_3_OH) 0.53; IR (cm^−1^): 3119, 3032 (C-H arom.), 2951 (C-H aliph.), 1732, 1714 (C=O, ester), 1654 (C=O, keto), 1518, 1498, 1467, 1410, 1392 (aromatic and heteroaromatic ring), 1255, 1236, 1197, 1174, 1114 (C–O–C, ester), 617, 563 (C–Br); ^1^H NMR (500 MHz, CDCl_3_): δ 9.54 (1H, d, *J* = 8.5 Hz, H-10), 8.79 (1H, s, H-6), 7.93-7.96 (2H, m, H-7, H-9), 7.82 (1H, dd, *J* = 7.0, 7.5 Hz, H-8), 7.69 (1H, s, CH of dibromoacetyl group from 3 position), 4.03 (3H, s, CH_3_ of methoxycarbonyl group from 1 position), 3.95 (3H, s, CH_3_ of methoxycarbonyl group from 2 position); ^13^C NMR (125 MHz, CDCl_3_): δ 176.8 (CO keto group from 3 position), 165.6 (CO keto ester from 2 position), 163.6 (CO keto ester from 1 position), 147.3 (C-6), 134.1 (C-9), 130.7 (C-8), 130.3 (C-3), 129.0 (C-10b), 128.3 (C-7),127.4 (C-10), 126.6 (C-10a), 121.8 (C-6a), 121.2 (C-2), 108.4 (C-1), 53.3 (CH_3_ of methoxycarbonyl group from 2 position), 52.7 (CH_3_ of methoxycarbonyl group from 1 position), 43.6 (CH of dibromoacetyl group from 3 position).

## 4. Conclusions

In conclusion, we report herein an efficient and straightforward pathway for obtaining a new class of blue fluorescent pyrrolodiazine derivatives under conventional (thermal) heating as well as under unconventional (microwave irradiation) heating. Blue fluorescent pyrrolodiazine derivatives have been obtained using a cycloaddition reaction of pyridazinium/phthalazinium ylides with activated alkynes. Under microwave irradiation, we isolated only fully aromatized cycloadducts with an increased yield (10–15%). The absorption and emission maxima and corresponding quantum yield of the obtained pyrrolodiazine derivatives were studied, with some of these compounds being moderate blue emitters. The fluorescence quantum yields of pyrrolodiazine derivatives are dramatically dependent on their structure, with only compounds with a pyrrolo-pyridazine skeleton having relative good quantum yields, whereas dihydropyrrolo-phthalazines show negligible and red shifted fluorescence emission.

In order to derivatizate these fluorescent pyrrolodiazines, we performed their bromination in heterogeneous catalysis using copper (II) bromide in chloroform/ethyl acetate. The reactions occur regioselectively both under conventional and unconventional heating to form only α-brominated products with increased reactivity. This increased reactivity should allow one to label various macromolecular structures of biological interest with fluorescent pyrrolodiazine moieties.

We also note that under MW irradiation, the reactions occur with increased selectivity regarding monobrominated compounds and offer several advantages in terms of yield, easier workup of the reaction, substantial decrease in consumed solvent and a substantial reduction in reaction time (from hours to minutes, and thus a consequent diminution in energy consumption). Taking into consideration these advantages, the proposed method should be considered environmentally friendly.

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
