# Peer review of "Microwave Assisted Reactions of Fluorescent Pyrrolodiazine Building Blocks"

_molecules, 2019, doi:10.3390/molecules24203760_

Round 1
Reviewer 1 Report
This manuscript by Costel Moldoveanu et al. reported the synthesis of 5-azaindolizine derivatives utilizing the 1,3-dipolar cycloaddition of N-alkylated pyridazinium/phthalazinium ylides with activated alkynes. The authors claimed that these fused nitrogen-containing compounds are of particular interest owing to their optical properties as intense blue emitters and therefore could be used as fluorescent biomarkers.
However, there is a lack of sufficient evidence in the manuscript to claim the use of these compounds as fluorescent biomarkers. Meanwhile, the microwave-assisted synthesis of the 5-azaindolizine derivatives reported in the manuscript does not add much value or prove to be advantageous than the already reported similar work by Dumitrascu and co-workers; wherein the synthesis of pyrrolophthalazine derivatives was described under conventional conditions.
The manuscript also needs improvement in terms of writing and figure presentation.
Author Response
This manuscript by Costel Moldoveanu et al. reported the synthesis of 5-azaindolizine derivatives utilizing the 1,3-dipolar cycloaddition of N-alkylated pyridazinium/phthalazinium ylides with activated alkynes. The authors claimed that these fused nitrogen-containing compounds are of particular interest owing to their optical properties as intense blue emitters and therefore could be used as fluorescent biomarkers.
However, there is a lack of sufficient evidence in the manuscript to claim the use of these compounds as fluorescent biomarkers.
We have modified the title of the article according with academic editor suggestion and removed the "potential fluorescent biological markers".
Meanwhile, the microwave-assisted synthesis of the 5-azaindolizine derivatives reported in the manuscript does not add much value or prove to be advantageous than the already reported similar work by Dumitrascu and co-workers; wherein the synthesis of pyrrolophthalazine derivatives was described under conventional conditions.
We note that under MW irradiation, the reactions occur with increased selectivity regarding to monobrominated compound and offers several advantages in terms of yield, easier workup of the reaction, substantial decrease in consumed, substantial reduction in reaction time (from hours to minutes) thus consequent diminution in energy consumption. Taking into consideration these advantages, the proposed method should be considered environmentally friendly. The new obtained mono- and di- brominated derivatives are new and never reported in the literature.
The manuscript also needs improvement in terms of writing and figure presentation.
We have modified the manuscript according with reviewer 4 suggestion and we were helped by a colleague fluent in English to improve the language style of the article.
Reviewer 2 Report
Dear Editor;
I have made careful evaluation of the manuscript "Microwave assisted reactions of pyrrolodiazine compounds as potential fluorescent biological markers" and my opinion is positive. Also there are a number of issues to which attention must be drawn.
Compound 8 is already a synthesized compound. Related literature is attached. It should be corrected.
Best Regards

Author Response
I have made careful evaluation of the manuscript "Microwave assisted reactions of pyrrolodiazine compounds as potential fluorescent biological markers" and my opinion is positive. Also there are a number of issues to which attention must be drawn.
Compound 8 is already a synthesized compound. Related literature is attached. It should be corrected.
We have modified the manuscript according with your suggestion highlighting better what was previously made in the literature and what brings new this paper. (see lines 73-81 from the revised manuscript).
Thank you for your suggestions.
Reviewer 3 Report
The authors presented a very interesting concept of synthesis and 5-azaindolizine derivatives by 1,3-dipolar cycloaddition reactions between N-alkylated pyridazine and methylpropiolate or dimethyl acetylenedicarboxylate (DMAD) by conventional reaction and in the presence of microwave radiation. Overall the study was very good. There was nothing major that I felt needed comment. The methods, results and discussions were all sufficient and clear.
I would suggest only a clarification in the title of the name of the compounds from "pyrrolodiazine 
compounds " to " 5-azaindolizine derivatives". Compound 7 is known in the literature (Masaki et al.[Chemical and Pharmaceutical Bulletin, 1973, vol. 21, p. 2780,2781]), and it would be worth mentioning this in the text. In the conclusions, it would also be worth adding a note about the potential use of the synthesized compounds. I feel that, after these corrections, the article is sufficient for use as-is.
Author Response
The authors presented a very interesting concept of synthesis and 5-azaindolizine derivatives by 1,3-dipolar cycloaddition reactions between N-alkylated pyridazine and methylpropiolate or dimethyl acetylenedicarboxylate (DMAD) by conventional reaction and in the presence of microwave radiation. Overall the study was very good. There was nothing major that I felt needed comment. The methods, results and discussions were all sufficient and clear.
I would suggest only a clarification in the title of the name of the compounds from "pyrrolodiazine 
compounds " to " 5-azaindolizine derivatives".
We have replaced "5-azaindolizine derivatives" with “pyrrolodiazine derivatives”.
Compound 7 is known in the literature (Masaki et al.[Chemical and Pharmaceutical Bulletin, 1973, vol. 21, p. 2780,2781]), and it would be worth mentioning this in the text.
We have modified the manuscript according with your suggestion highlighting better what was previously made in the literature and what brings new this paper. (see lines 73-81 from the revised manuscript).
In the conclusions, it would also be worth adding a note about the potential use of the synthesized compounds.
We have modified the manuscript according with your suggestion (see lines 362-368 and 375-377 from the revised manuscript).
I feel that, after these corrections, the article is sufficient for use as-is.
Thank you for all suggestions.
Reviewer 4 Report
This manuscript describes the synthesis and the examination of the photophysical properties of new blue fluorescent pyrrolodiazines. Their modification to reactive bromoketones is also investigated, in order to increase their reactivity for potential applications as fluorescent labels for bioconjugation. As such, the paper is well written and organised, and it fits well the scope of the journal Molecules. Therefore it deserves to be published after the following minor comments have been addressed:
Line 20: labels “7-10” should be bold
Line 44: superscript 2 in “sp2”
Line 47: sentence is incomplete
Line 48: “diazinium ylides”
Line 59 (and same on line 137): “by the reaction of acetone with”
Line 77: “cycloadducts”
Line 117: replace “enriched” with “enhanced”
Scheme 2: the usual shortened form for ethyl acetate is “EtOAc”, not EtAc, please correct
Line 163 and following: In the experimental section, the abbreviation “mMol” should not be used, and replaced with “mmol”.
Line 186: “dropwise”
Regarding the schemes, I feel that it is unnecessary to explicitly draw CH2, CH3 and CO groups, and the skeletal formulae should be used instead.
Furthermore, a different style has been used in the two schemes to identify the pyridazine/phthalazine series. In scheme 1 the numbers are red, in scheme 2 they are black and only the name is red. The author should choose one format and keep it consistent.
Author Response
This manuscript describes the synthesis and the examination of the photophysical properties of new blue fluorescent pyrrolodiazines. Their modification to reactive bromoketones is also investigated, in order to increase their reactivity for potential applications as fluorescent labels for bioconjugation. As such, the paper is well written and organised, and it fits well the scope of the journal Molecules.
Therefore it deserves to be published after the following minor comments have been addressed:
Line 20: labels “7-10” should be bold
We have modified the manuscript according with your suggestion.
Line 44: superscript 2 in “sp2”
We have modified the manuscript according with your suggestion.
Line 47: sentence is incomplete
We have modified the manuscript according with your suggestion.
Line 48: “diazinium ylides”
We have modified the manuscript according with your suggestion.
Line 59 (and same on line 137): “by the reaction of acetone with”
We have modified the manuscript according with your suggestion.
Line 77: “cycloadducts”
We have modified the manuscript according with your suggestion.
Line 117: replace “enriched” with “enhanced”
We have modified the manuscript according with your suggestion.
Scheme 2: the usual shortened form for ethyl acetate is “EtOAc”, not EtAc, please correct
We have modified the Scheme 2 according with your suggestion.
Line 163 and following: In the experimental section, the abbreviation “mMol” should not be used, and replaced with “mmol”.
We have modified the manuscript according with your suggestion.
Line 186: “dropwise”
We have modified the manuscript according with your suggestion.
Regarding the schemes, I feel that it is unnecessary to explicitly draw CH2, CH3 and CO groups, and the skeletal formulae should be used instead.
We have modified the schemes according with your suggestion.
Furthermore, a different style has been used in the two schemes to identify the pyridazine/phthalazine series. In scheme 1 the numbers are red, in scheme 2 they are black and only the name is red. The author should choose one format and keep it consistent.
We have modified the schemes according with your suggestion.
Thank you for your all suggestions.
Round 2
Reviewer 1 Report
This revised manuscript reads better than the previous version. Overall, the major highlight of this paper is that the use of microwave irradiation improved the yield of a chemical synthesis that could be conducted under conventional conditions. The improvement was incremental in my opinion, especially to the synthesis of relatively simple structures. The measurement of fluorescence properties of the synthesized pyrrolopyridazines, as presented in the current context, does not offer much insight or value with regards to how these compounds could be useful in a biological setting.